# Incorporation of Tapioca Starch and Wheat Flour on Physicochemical Properties and Sensory Attributes of Meat-Based Snacks from Beef Scraps

**DOI:** 10.3390/foods11071034

**Published:** 2022-04-02

**Authors:** Hataitip Nimitkeatkai, Kannika Pasada, Amnat Jarerat

**Affiliations:** 1School of Agriculture and Natural Resources, University of Phayao, Phayao 56000, Thailand; hataitip.ni@up.ac.th; 2Unit of Scientific Laboratory for Education, Kanchanaburi Campus, Mahidol University, Kanchanaburi 71150, Thailand; kannika.pas@mahidol.ac.th; 3Food Technology Program, Kanchanaburi Campus, Mahidol University, Kanchanaburi 71150, Thailand

**Keywords:** beef snacks, physicochemical property, sensory property, tapioca starch, wheat flour

## Abstract

The global demand for healthy snacks with high protein content is growing annually. Meat scraps generated after meat cutting in the slaughtering process are considered a valuable protein product. The aim of this research was to formulate the meat-based snacks obtained from beef scraps by baking at 150 °C for 20 min. The physicochemical properties, texture and sensory profiles of the beef snacks were investigated. Among tapioca starch, modified starch and wheat flour, the texture profiles and scanning electron microscopy (SEM) revealed that wheat flour contributed to a firm texture of the products, resulting in significantly (*p* < 0.05) higher sensory scores for texture. The overall acceptability based on physicochemical and sensory attributes of wheat flour were significantly (*p* < 0.05) higher than tapioca starch and modified starch. The results showed that the relatively low content of wheat flour at 0.625% (*w/w*) was of sufficient proportion to provide proper physicochemical properties and texture attributes to beef snacks. In addition, the results also indicated that the desirable properties of the obtained meat-based snacks were influenced by the type and content of starch and/or flour used. This study reveals the benefits of meat scraps as a potential protein-rich source and further applications in other meat-based snacks.

## 1. Introduction

Snacks have typically been associated with the consumption of products with low nutritional value. New dietary trends, on the other hand, have switched toward the consumption of protein-rich foods [1,2]. Meat snacks have been manufactured as high-quality products with a high significance as a convenient food product for people worldwide [3].

Beef is a rich source of macronutrients as well as micronutrients. The macronutrients of meat include high-value protein and fat, containing saturated and unsaturated fatty acids. The micronutrients in beef and edible offal include heme iron, zinc, selenium, and vitamins D, B1 (thiamine), B2 (riboflavin), B3 (niacin), B5 (pantothenic acid), B6 (pyridoxin), and B12 (cobalamin). All of these nutrients are important for human physiological functions [4]. The high concentration of nutrients in beef snacks, such as proteins, minerals (heme), and biologically active peptides (carnosine), has been reported [3].

A plant-based diet is specified as one that emphasizes foods derived from plants. Fruits, vegetables, grains, legumes, nuts, and meat substitutes, including soybean products, are all examples of this. The consumption of plant-based proteins has attracted attention in recent years [5]. Although meat has a high protein content and nutritional profile, plants have nutrients, such as fiber and flavonoids, that are lacking in animal foods. Therefore, eating balanced amounts of both is the best way to get all the nutrients more efficiently. It was reported that consumption of a high-protein snack (77%) as an afternoon snack delayed the request for dinner by 60 min. In contrast, high-fat snacks (58% of energy from fat) prolonged dinner requests by 25 min, whereas high-carbohydrate snacks (84%) delayed dinner requests by 34 min [6]. This implies that the relatively high dietary energy density of high-protein snacks increases the feeling of fullness.

Non-meat components, including starches and flour, can greatly increase the production of better-quality meat products [7]. Starches and flours are plant products with high nutritional components, including cereals, such as wheat and maize, and also root crops, such as tapioca [8]. In food applications, starch and flour are multifunctional food ingredients and have been used as a texturizer, a binder, an emulsion stabilizer, a moisture retainer or a coating agent and are relatively affordable [9]. Tapioca starch contains about 17% amylose, which contributes as a good binder, by absorbing water and forming a thick gel with high viscosity, improving water/moisture retention and cooking yield. Wheat flour, on the other hand, contains a higher percentage of amylose (20% to 25%) and protein (8% to 15%), which provides products a better expansion quality in cooking [10].

Scraps of meat are generally obtained from slaughtering after the manual removal of small pieces of meat left on the bone. An estimation of the market beef carcass generates 25% bone-in trimmed steaks, 25% bone-in roasts, 25% stew beef and ground beef, both boneless, and 25% wastage. The whole carcass accounts for 63% of the market live weight of cattle [11]. The efficient utilization of low commercial value animal by-products as a source of cheaper and nutrient-dense food ingredients is of considerable interest [12]. The application of producing a protein-based snack obtained from meat scraps could provide an approach towards increasing their value and utilization at a reasonable cost, which motivated us to focus our attention on this for the study.

Many studies in the literature have focused on the use of starch and flour as the components, in high amounts (30–70%), for meat-based snacks [13,14,15,16]. However, there is no such report that relates the incorporation of relatively low starch and flour content to the physiochemical properties of beef snacks. Therefore, to obtain the better textural and sensory qualities of the potential protein-based product of meat, the objective of this research was to formulate the protein-rich snack by the baking process. The influence of tapioca starch, modified starch and wheat flour in a relatively low content (0.625–5%) on the physicochemical, textural, and sensory qualities of beef snacks obtained from beef scraps was also elucidated.

## 2. Materials and Methods

### 2.1. Preparation of Beef Snack Chip

In this study, beef snacks were prepared according to the modified formula of our laboratory. The formula is presented in Table 1. The ingredients were: beef scraps (purchased from a local slaughterhouse in Phayao city, Thailand), iced water, sugar, soy sauce, spice blends, and sodium bicarbonate.

The experiments were divided into 2 parts: (i) the impact of the type of starch/flour incorporated at a concentration of 2.5%, and (ii) the impact of wheat flour concentration (between 0 and 5%) on product properties.

For the influence of starch/flour type, the following ingredients of flour or starch were added to the recipe at 2.5% compared to 0% (control): tapioca starch (Bangkok Inter Food Co., Ltd., Bangkok, Thailand), wheat flour (10.5–11.0% protein, United Flour Mill Public Co., Ltd., Bangkok, Thailand), pregelatinized tapioca starch (BINDGEL^®^, Siam Modified Starch Co., Ltd., Pathum Thani, Thailand), or a mixture of tapioca starch and wheat flour (1:1).

For the influence of the wheat flour incorporation experiment, 0.625, 1.25, 2.5, and 5% of wheat flour were added to the recipe by substituting the concentration of beef scraps with the wheat flour content, while the rest of the ingredients were the same in all batches.

The meat scraps were ground and mixed with all of the ingredients in a Philips HR7761 Food Processor (Philips, Amsterdam, The Netherlands) until a uniform batter was obtained. The batter was then spread on a plastic sheet and cured at 4 °C for 12 h. The sheet of batter was dried at 55 °C for 90 min in a tray drier (FnB Machinery & Solution, Co. Ltd., Bangkok, Thailand) and cut into square-shaped pieces (2.5 × 2.5 cm) with a thickness of 0.2 cm by using cooking scissors. The dry samples were baked at 150 °C for 20 min in a Sharp EO-42K Economic Deck Oven (Federal Electric Co., Ltd., Samut Prakan, Thailand). The finished products were sealed in polyethylene zip bags and stored at ambient temperature (approximately 28 °C to 32 °C) for subsequent measurement of physicochemical properties and sensory evaluation. The production of the beef snacks was conducted in triplicate (three independent experiments).

### 2.2. Analysis Methods

#### 2.2.1. Measurement of Water Activity (a_w_)

Samples from each treatment were selected and cut into small pieces with sharp scissors, and 1 g of each sample was placed in water activity cups. Water activity, represented as the free water level of the sample, was determined with a water activity meter (Aqualab-4TE, Pullman, WA, USA). Water activity was measured in triplicate.

#### 2.2.2. Measurement of Moisture Content

Five grams of sample were weighted and dried at 105 °C until the sample reached equilibrium in a hot air oven (FD 115, Binder, Tuttlingen, Germany). Afterward, the samples were weighed, and the percentage of moisture content was calculated. The moisture content was determined in triplicate on each snack product.

#### 2.2.3. Measurement of Color

The color of the samples was analyzed using a colorimeter Hunter LAB (ColorQuest XE, Hunter Lab, Reston, VA, USA). The CIELab color values were used with *L** (lightness), *a** (tendency towards green −, or red +) and *b** (tendency towards blue −, or yellow +). All samples were read at two different positions and measured in triplicate.

#### 2.2.4. Measurement of Texture Profile

Textural attributes were analyzed by a TA-XT plus Texture Analyzer (Stable Microsystems, Godalming, UK) with a crisp fracture support rig, force-versus-distance compression program. The texture profile analysis settings were as follows: Pre-Test Speed: 1.0 mm/s; Test Speed: 1.0 mm/s; Post-Test Speed: 10.0 mm/s; Strain: 70%; Time: 5.00 s; Trigger Force (auto): 0.05 N. Hardness and fracturability values were measured in each test. The hardness was determined from the maximum peak force in newtons (N), fracturability was determined from the distance (mm) at which the sample breaks. Five texture measurements were conducted for each snack product and were performed in triplicate.

### 2.3. Sensory Evaluation

Sensory evaluation of samples was carried out by 30 untrained panelists (15 males and 15 females, aged 18–38 years old), undergraduate and graduate students of the School of Agriculture and Natural Resources, University of Phayao, Thailand. Before starting the analysis, all participants (volunteers) in this experiment were informed of the aim of the sensory evaluation, and gave their informed consent to perform the evaluation. The panelists evaluated the samples using a 7-point hedonic scale ranging from 1 (dislike extremely) to 7 (like extremely), indicating very low to very high desirability for color, flavor, texture and overall acceptability. A glass of water is also provided to cleanse the palate.

### 2.4. Microstructure Evaluation

Beef snacks (cross-sectional pieces) were freeze dried. Double sticky tape was used to mount the beef snacks on aluminum stubs. The samples were sputter-coated with platinum to render them thermoelectrically conductive by using a JEC-3000 FC auto fine coater (JEOL, Tokyo, Japan) and then scanned using a JSM-IT200 scanning electron microscope (JEOL, Tokyo, Japan). The study was carried out with a 20 kV acceleration voltage to avoid damage to a sample caused by power beams. The micrographs were taken at a magnification of 85×.

### 2.5. Statistical Analysis

The experiments were conducted using a completely randomized design (CRD). Statistical analysis of the recorded data was performed using SPSS v.18.0 (SPSS Inc., Chicago, IL, USA). One-way ANOVA and Duncan’s multiple range tests were carried out to determine the variance between the treatments. The results were presented as the mean ± standard deviation (SD), and statistical significance was expressed at a level of *p* < 0.05.

## 3. Results

### 3.1. Effect of Starch or Flour Type on the Physicochemical Properties and Sensory Evaluation of Beef Snacks

The influence of the type of starch/flour on the desirable properties of the obtained meat-based snacks from beef scraps was investigated, according to the formulation indicated in Table 1. In order to study the possible use of native ingredients in beef snacks, the combination of tapioca starch and wheat flour was also examined. The visual appearance of the different formulations is shown in Figure 1. The incorporation of tapioca starch increased the darkness considerably, making it noticeable in the appearance of beef snacks.

As shown in Table 2, beef snacks formulated with tapioca + wheat and wheat flour provided higher lightness (*L**) than the control and tapioca starch (*p* < 0.05). The results of *L** also showed the same tendency as the visual appearance. All beef snacks formulated with starch or flour showed a higher redness (*a**) than control. The yellowness (*b**) of beef snacks incorporated with wheat flour was significantly increased, followed by tapioca + wheat and modified starch, respectively (*p* < 0.05). There were no significant differences between tapioca starch and control on *L** and *b**.

The physicochemical and textural qualities of beef snacks are shown in Table 2. When compared to control, beef snacks contained lower moisture content and water activity after being incorporated with starch or flour. The ranges in moisture content and water activity were between 2.40–3.36% and 0.44–0.51, respectively. However, the moisture content and water activity of starch or flours formulated at 2.5% (*w*/*w*) exhibited no significant difference with the control.

As shown in Table 2, the hardness of the beef snacks ranged from 558.38 to 1030.57 N, which revealed that the product formulated with modified starch resulted in the highest hardness, significantly (*p* < 0.05), followed by the control, tapioca starch, wheat flour and tapioca + wheat, respectively. The products obtained from the combination of tapioca and wheat flour provided light and slightly yellow products, with a relatively low value of hardness (Table 2). On the other hand, regarding the results of fracturability, there was no statistically significant differences observed among wheat flour and starches.

The color of beef snacks formulated with modified starch, wheat flour and tapioca + wheat was more appreciated by consumers than that formulated with tapioca starch and control (Table 3). The results of sensory evaluation were in the same tendency with the values of *L**, *a** and *b**, shown in Table 2. The average scores of overall acceptability ranged from 3.87 to 5.83 in the 7-point hedonic scale. We observed that higher color and texture scores were related to a better appreciation by consumers. Beef snacks formulated with wheat flour and modified starch were the preferred samples (*p* < 0.05), with overall acceptability scores of 5.83 and 5.80, respectively (Table 3). Additionally, beef snacks formulated with wheat flour exhibited a significantly (*p* < 0.05) higher crispness than other formulations.

Figure 2 shows the micrographs of beef snacks made from different starch/flour. As shown in Figure 2d, SEM observation revealed that beef snacks incorporated with wheat flour had irregular holes and pits covering the cross-sectional area, which was also evidenced by significantly low hardness values of 570.89 N (Table 2). In contrast, control without starch/flour had a compact structure (Figure 2a), which also corresponded with the relatively high hardness (903.87 N). The similarity of microstructures was observed in the beef snacks incorporated with modified starch (Figure 2c) , wheat flour (Figure 2d) and tapioca starch + wheat flour (Figure 2e). However, the morphology of the beef snacks formulated with starch and flour was much better than control, in terms of the porous structure and hole appearance.

It was found that the addition of wheat flour to beef snacks provided the most preferences. In the following part, wheat flour was selected to formulate and produce beef snacks, using the following amounts: 0.625, 1.25, 2.5, and 5.00% (*w/w*).

### 3.2. Effect of Wheat Flour Content on the Physicochemical Properties and Sensory Evaluation of Beef Snacks

The influence of the amount of wheat flour on the desirable properties of the obtained meat-based snack chips was further investigated. As shown in Table 4, beef snacks formulated with wheat flour provided higher lightness (*L**), redness (*a**) and yellowness (*b**) than the control (*p* < 0.05). There were no significant differences in the color of the beef snacks formulated with wheat flour in the range 0.625 to 5.00 (*w/w*).

Figure 3 shows that wheat flour content in the range of 0–5% (*w*/*w*) had an effect on moisture content and water activity of beef snack samples (Figure 3a). The moisture content and the water activity were decreased in a similar tendency to the increase in wheat flour content. The increase in wheat flour content in the samples also resulted in the tendency to decrease in hardness and fracturability (Figure 3b).

Figure 4 shows the microstructure of control and beef snacks formulated from different wheat flour contents, ranging from 0.625 to 5.00% (*w/w*). SEM observation revealed that increasing wheat flour content led to an increase in the visible holes and pits, which corresponded to the decrease in hardness (Figure 3b).

The average score of overall acceptability ranged from 3.87 to 5.63 on the 7-point hedonic scale (Table 5). Formulation with the relatively low concentration of wheat flour at 0.625–2.5% (*w/w*) exhibited more appreciated color and overall acceptability (*p* < 0.05) than that of higher proportion (5% *w/w*). As for the obtained results on the color and overall acceptability, beef snacks incorporated with 0.625% (*w/w*) wheat flour were the preferred samples. Overall acceptability of the beef snacks formulated with 0.625% (*w/w*) wheat flour was the highest amongst the obtained samples.

## 4. Discussion

From the results, the color of beef snacks was influenced by starch or flour type, as well as the flour content (Table 2 and Table 4). When starch or flour was added, the products became lighter, redder, and yellower compared with the control. Beef snacks incorporated with flour and starch samples had yellowness coloration and a more reddish color (higher *a** value) than the control. This occurrence may be related to the moderate production of browning compounds formed in the Maillard reactions due to interactions between protein and carbohydrate at high temperatures [17]. Color influences the quality of food and its appeal to consumers. When most foods are heated, the Maillard reaction is primarily responsible for color development [18]. In addition, the baking process also develops flavor and aroma due to the browning effect of the Maillard reaction on the food surface [19]. These occurrences provided the most positive effects on the sensory attributes of the beef snacks obtained from wheat flour with 0.625% (*w*/*w*) (Table 3 and Table 5).

Wheat flour is the only cereal flour that can form cohesive dough upon hydration, hence, it is commonly used to make baking products, including bread [20]. Microstructure observation by SEM revealed that beef snacks incorporated with wheat flour contained a relatively large number of pits and holes compared to tapioca starch, modified starch and control (Figure 2). A similarity was observed between using wheat flour and wheat flour + tapioca starch. On the other hand, flat and dense areas appeared between the pits and holes of the prepared beef snacks using modified tapioca starch (Figure 2c), which corresponded to the relatively high value of hardness (Table 2). According to Peighambardoust et al. [21], a large number of holes and pits were due to gas entrapment during the batter preparation. These properties were observed in this study, mainly attributed to the gluten proteins that generated a continuous viscoelastic network and, thereby, entrapped gas [22]. As reported by Shewry et al. [23], protein constitutes 7% to 15% of typical wheat flour, including albumins and globulins, while gluten accounts for 80% to 90% of flour proteins.

For the results of beef snacks formulated using tapioca starch, an amount of 2.5% (*w*/*w*) led to a homogenous matrix between the meat protein and starch granules (Figure 2b). The dense structure was observed, possibly due to the starch retrogradation [24]. The modified starch used in this study was the tapioca starch that had already broken down the intermolecular bonds of starch molecules in the presence of water and heat, allowing the hydrogen bonding sites to engage more water and enhance swelling power [25]. Its features provided the homogenous matrix of the products, which accounted for the flat areas with some porous structures after the baking process (Figure 2c). The significantly highest hardness value of beef snacks incorporated with pregelatinized starch (*p* < 0.05) may be due to the rearrangement and association of the dispersed amylose short chains through hydrogen bonding, so called starch retrogradation [24]. Wheat flour could probably induce a more uniform and viscoelastic protein matrix compared to cassava and modified starch. However, it is complicated to correlate a precise alteration to a specific effect of these ingredients in the starch–meat complex in the current study.

As observed in Figure 4, the number and size of fine structures and pores increased with increasing wheat flour percentage, the product structure became weak to withstand the compressive force, hardness and fracturability tended to decline (Figure 3b). The presence of a higher amount of gluten also provided crispiness, texture, and nutritive value to snacks, which was consistent with a previous report by Kumar et al. [10]. Based on SEM observation, it was noticeable that there was no appearance of starch granules in the sample matrix, implying a homogeneous structure was obtained.

As protein is the main ingredient in meat products, interactions between protein and carbohydrates can play a major role in the functional characteristics of the end product [26]. Formulation using wheat flour affected the texture of beef snacks by strengthening the starch–protein matrix and led to the most significant highest crispiness and other sensory attributes amongst starches and flour (Table 3). According to the results on the incorporation of wheat flour at 2.5%, all batches were within the same range of physicochemical properties (Table 2 and Table 4, Figure 3a,b). Additionally, increasing the wheat flour percentage had no effect on sensory attributes and overall acceptability (Table 5).

The main purpose of lowering water activity in food is to prevent or reduce microbial growth. The minimum for all microbial growth is 0.60, and food spoilage due to microbial growth would not occur below this value [27]. The water activity values of the samples in the present study ranged from 0.41 to 0.49, which were insufficient conditions for microbial growth (Table 2 and Figure 3a). Similar studies reported that the moisture retention of meat products was due to the ability of the protein matrix to retain water [28]. The present study revealed that the different moisture retention in all products (2.4–3.3%), including the control, can be due to the different level of protein content and its ability to keep moisture in the snack matrix. According to the results, water activity in the control was significantly higher than in other formulations (Table 2 and Figure 3a). It can also be observed that an increase in wheat flour content produced a decrease in hardness (Figure 3b). The decrease in hardness was mainly due to the reduction in moisture content in the beef snacks matrix after baking. Additionally, a proper wheat flour content could ensure the microbial safety of the obtained beef snacks by reducing available water (water activity) for microbial growth.

In addition to wheat protein, the textural properties of products can be altered because of the characteristics of starch granules in absorbing water and expanding themselves during heating. The starch granules produce more reinforcement in the gel matrix after they absorb water and swell [29]. Amylose within flour or starch could strengthen the dough, which, in turn, improves the forming and cutting properties of dough to produce snack foods with a crunchy texture [30]. Research from Oladunmoye et al. [31] showed that the amylose content of the regular wheat flour was 28.19%, whereas tapioca starch had lower amylose content with 19.49%. In this study, wheat flour containing a relatively high amount of amylose is recommended in beef snack food formulations, to obtain products with a crunchy texture.

Textural properties are of high interest in terms of consumer acceptance. Regarding snacks, hardness and fracturability are considered important parameters [15]. Hardness is defined as the force required to achieve a given deformation of crispy chips. The beef snacks fractured more easily with increasing wheat flour content; thus, the protein domain would become less rigid. Low hardness resulted from a loose matrix caused by their porosity, while high hardness altered the firmness of beef snacks. Consequently, the hardness value of the incorporation with 0.625% (*w*/*w*) of wheat flour was within the appropriate range and can be considered as a preference by sensory evaluation.

## 5. Conclusions

The results obtained from this study indicated that wheat flour, when compared to tapioca starch and modified tapioca starch, was the most potent composition for beef snack processing. The texture profiles also indicated that products made with wheat flour showed a relatively firmer texture. The sensory panelists also preferred products made with wheat flour when compared to products made with tapioca and modified starches, on the appearance, flavor, texture, and overall acceptability. The relatively low content of wheat flour at 0.625% (*w*/*w*) was considerable enough to be sufficient to provide crispiness and proper texture to beef snacks with the lower water activity and moisture content. Among the starch and flour used in this study, wheat flour containing a higher protein content (10.5–11%) imparted products with better physicochemical properties and sensory attributes, while the incorporation with a higher amount of wheat flour content attributed low hardness to texture. We have demonstrated that the incorporation of the most commonly used non-meat ingredients, including wheat flour, into beef snacks increased the positive impact on organoleptic attributes of this meat-based product, which can be considered as the greatest choice, in terms of increasing the utilization of meat scraps for the snack food industry.

## Figures and Tables

**Figure 1 foods-11-01034-f001:**
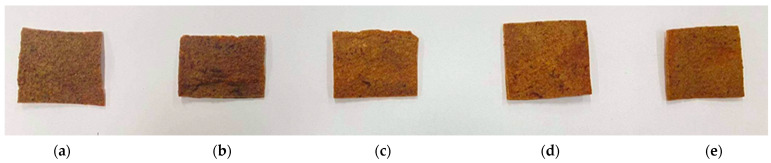
Visual appearance of beef snacks formulated without starch or flour (control) (**a**); with tapioca starch (**b**); with modified starch (**c**); with wheat flour (**d**) and with tapioca starch plus wheat flour (**e**).

**Figure 2 foods-11-01034-f002:**
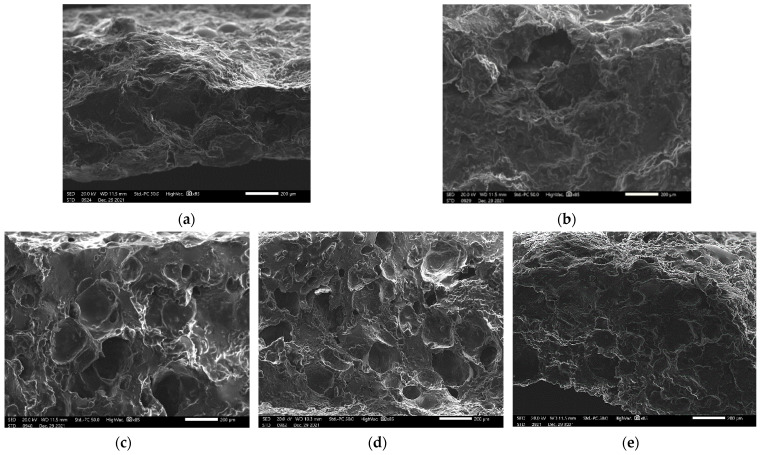
Cross-section microstructure of beef snacks without incorporation of starch/flour (**a**), with tapioca starch (**b**), with pregelatinized starch (**c**), with wheat flour (**d**) and with tapioca starch + wheat flour (**e**) at 85× magnification. Bar = 200 μm.

**Figure 3 foods-11-01034-f003:**
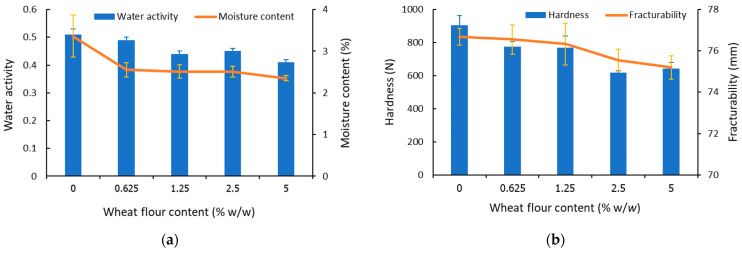
Water activity and moisture content (**a**); hardness and fracturability (**b**) of beef snacks formulated with different wheat flour content.

**Figure 4 foods-11-01034-f004:**
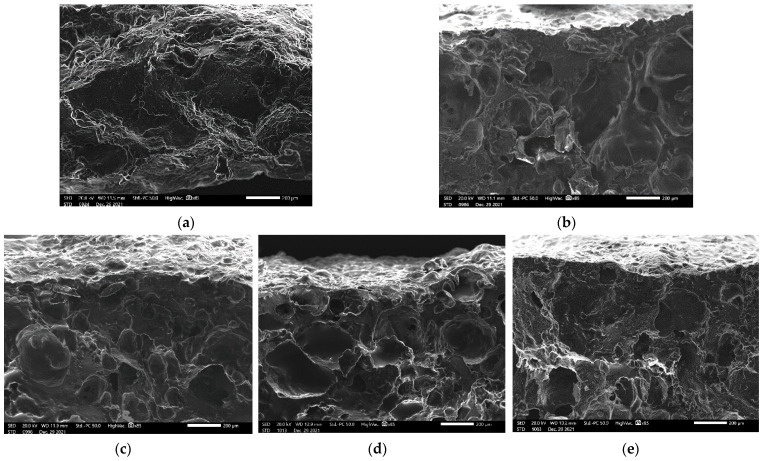
Cross-section microstructure of beef snacks without incorporation of wheat flour (**a**), 0.625% (*w/w*) (**b**), 1.25% (*w/w*) (**c**), 2.50% (*w/w*) (**d**) and 5.0% (*w/w*) (**e**) wheat flour at 85× magnification. Bar = 200 μm.

**Table 1 foods-11-01034-t001:** The formula of beef snacks with different starch/flour type.

Ingredients	Concentration (%, *w*/*w*)
Beef scraps	77.00–79.50
Sugar	9.50
Soy sauce	7.50
Coriander powder, cumin powder, ground black pepper	1.50
Table salt	1.00
Sodium bicarbonate	1.00
Starch/flour ^1^	0.00–2.50

^1^ Starch and/or flour formulation: addition of 2.5% concentration of tapioca starch, wheat flour, modified starch or tapioca starch + wheat flour based on raw meat scraps weight (*w*/*w*).

**Table 2 foods-11-01034-t002:** Physicochemical properties and texture of beef snacks formulated with starch or flours.

Physical Properties	Control	Tapioca Starch	Modified Starch	Wheat Flour	Tapioca Starch + Wheat Flour
Lightness (*L**)	19.16 ± 1.98 ^b^	19.34 ± 1.70 ^b^	22.22 ± 2.37 ^ab^	23.80 ± 2.82 ^a^	25.04 ± 1.52 ^a^
Redness (*a**)	7.37 ± 0.72 ^b^	9.43 ± 1.48 ^a^	10.30 ± 0.78 ^a^	10.67 ± 0.76 ^a^	9.17 ± 0.65 ^a^
Yellowness (*b**)	14.67 ± 0.16 ^c^	16.06 ± 0.79 ^c^	19.86 ± 1.38 ^b^	21.63 ± 0.67 ^a^	20.23 ± 0.11 ^ab^
Moisture content (%)	3.36 ± 0.75	2.40 ± 0.05	2.98 ± 0.07	2.84 ± 0.13	2.72 ± 0.10
Water activity	0.51 ± 0.02	0.44 ± 0.01	0.48 ± 0.01	0.45 ± 0.01	0.46 ± 0.01
Hardness (N)	903.87 ± 80.57 ^b^	734.46 ± 55.53 ^c^	1030.57 ± 0.39 ^a^	570.89 ± 75.60 ^d^	558.36 ± 40.15 ^d^
Fracturability (mm)	76.67 ± 0.41	76.07 ± 0.20	75.50 ± 0.31	75.75 ± 0.89	75.98 ± 0.15

^a–d^ Means ± SD (*n* = 3) in the row with different small superscript letters indicate significant difference at *p* < 0.05 level.

**Table 3 foods-11-01034-t003:** Sensory evaluation * of beef snack chips formulated with starch or flours.

Formulations	Color	Flavor	Crispness	Overall Acceptability
Control	3.50 ± 1.50 ^c^	4.20 ± 1.60 ^c^	3.33 ± 2.07 ^c^	3.87 ± 1.57 ^c^
Tapioca starch	4.23 ± 1.35 ^b^	5.00 ± 1.29 ^b^	4.90 ± 1.24 ^b^	5.10 ± 1.21 ^b^
Modified starch	5.33 ± 1.45 ^a^	5.30 ± 1.09 ^ab^	5.80 ± 1.00 ^a^	5.80 ± 1.13 ^a^
Wheat flour	5.50 ± 1.25 ^a^	5.73 ± 0.98 ^a^	6.13 ± 1.07 ^a^	5.83 ± 1.02 ^a^
Tapioca starch + Wheat flour	5.57 ± 1.28 ^a^	5.23 ± 1.55 ^ab^	5.60 ± 1.50 ^ab^	5.63 ± 1.45 ^ab^

* Based on a 7-point hedonic scale (1 = dislike extremely; 7 = like extremely). ^a–c^ Means ± SD in the row with different small superscript letters indicate significant difference at *p* < 0.05 level.

**Table 4 foods-11-01034-t004:** Color (CIE *L**, *a**, *b**) of beef snacks formulated with different wheat flour contents.

Wheat Flour (%*w/w*)	Lightness (*L**)	Redness (*a**)	Yellowness (*b**)
0	19.16 ± 1.98 ^b^	7.37 ± 0.72 ^b^	14.67 ± 0.16 ^b^
0.625	26.30 ± 0.95 ^a^	9.50 ± 1.15 ^a^	17.94 ± 0.38 ^a^
1.25	27.97 ± 0.86 ^a^	9.88 ± 1.33 ^a^	19.15 ± 0.57 ^a^
2.50	28.75 ± 2.44 ^a^	10.62 ± 0.60 ^a^	19.62 ± 0.72 ^a^
5.00	28.43 ± 0.14 ^a^	11.12 ± 0.31 ^a^	19.88 ± 1.31 ^a^

^a,b^ Means ± SD (*n* = 3) in the column with different small superscript letters indicate significant difference at *p* < 0.05 level.

**Table 5 foods-11-01034-t005:** Sensory evaluation * of beef snacks formulated with different wheat flour contents.

Wheat Flour (%*w/w*)	Color	Flavor	Crispness	Overall Acceptability
0	3.50 ± 1.50 ^c^	4.20 ± 1.60 ^b^	3.33 ± 2.07 ^b^	3.87 ± 1.57 ^c^
0.625	5.27 ± 1.36 ^a^	5.53 ± 1.20 ^a^	5.47 ± 1.20 ^a^	5.63 ± 1.32 ^a^
1.25	5.27 ± 1.39 ^a^	5.27 ± 1.55 ^a^	5.27 ± 1.34 ^a^	5.13 ± 1.80 ^ab^
2.50	5.10 ± 1.45 ^ab^	5.33 ± 1.45 ^a^	5.00 ± 1.51 ^a^	5.00 ± 1.60 ^ab^
5.00	5.67 ± 1.67 ^b^	5.07 ± 1.62 ^a^	4.93 ± 1.64 ^a^	4.77 ± 1.65 ^b^

* Based on a 7-point hedonic scale (1 = dislike extremely; 7 = like extremely). ^a–c^ Means ± SD in the column with different small superscript letters indicate significant difference at *p* < 0.05 level.

## Data Availability

Data are contained within the article.

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
