# Peer review of "Incorporation of Tapioca Starch and Wheat Flour on Physicochemical Properties and Sensory Attributes of Meat-Based Snacks from Beef Scraps"

_foods, 2022, doi:10.3390/foods11071034_

Round 1

Reviewer 1 Report

Based on physicochemical, texture and sensory studies, the objective of the present study was to formulate a protein-rich baked snack that combines animal (i.e. beef scraps) and plant (tapioca starch, wheat flour or modified starch) sources. However, the modifications listed below should be made:

  • Page 1/Line 32: What does “occupied food product” mean?
  • Page 2/Section 2.1: The authors should describe in more detail all the snack formulations they have prepared, making it clear that two production runs have been made: i) a first to test the impact of the type of plant sources added at a rate of 2.5% and ii) a second to test, in the case of wheat flour, the impact of the added rate (between 0 and 5%) on product properties. As a result, Table 1 will also need to be modified to best reflect the above comment.
  • Page 3/Sections 2.2.1 & 2.2.2: Please indicate precisely the number of samples for each of the analyses mentioned in these sections.
  • Page 3/Line 109: Was the humidity measurement carried out according to a standard? If so, which one? How were the 5 hours of drying determined?
  • Page 3/Section 2.2.3: The way in which the colour measurement was carried out should be described in much greater detail, e.g. by indicating the illuminator, the angle, etc.
  • Page 3/Line 125: The age range of people recruited for sensory analysis is very narrow. Why this choice? Because of the target audience for this type of food? Not really a choice? If not, could this not introduce a bias in the sensory study with respect to the general population likely to consume this type of food? Please explain.
  • Page 3/Section 2.2.4: you should clearly define here how you actually assess the hardness and fracturability of the different samples analysed, the results of which are presented and discussed in sections 3 & 4.
  • Page 4/Figure 1: Please add the letters a-e next to the images so that readers can unambiguously link the images to the figure legend.
  • Page 4/Table 2: Please indicate, for each measured quantity, the number of samples analysed, i.e. the number of replicates.
  • Page 5/Lines 180-181: There must be a word (or words) missing from this sentence. Read it again carefully.
  • Page 8/Section 4: In this section devoted to the discussion of the results, I am convinced that it would have been interesting to compare and discuss the results of the column 'Wheat flour' in Table 2 with those of the fourth row of Table 4 and the abscissa '2.5' in Figures 3a & 3b, which, if I am not mistaken, correspond, in a way, to repeats.

Finally, although I am not an English speaker, I would advise authors to have their manuscript carefully proofread by an English-speaking person, as some turns of phrase seem to me to have been clumsily written.

Reviewer 2 Report

The objective of the manuscript is not clear enough.The title does not reflect the content.

What is the relationship between Tapioca starch and wheat flour? The link must be disclosed.

On the other hand, how many times was the production repeated?More detailed information must be given for statistical analysis.

Round 2

Reviewer 2 Report

Line 128-Was the color of samples measured using a colorimeter or spectrophotometer? Please check.